# Identifying topological corner states in two-dimensional metal-organic frameworks

Tianyi Hu [1], Weiliang Zhong[2], Tingfeng Zhang [1], Weihua Wang [2] ✉ & Z. F. Wang [1,3] ✉

Due to the diversity of molecular building blocks, the two-dimensional (2D) metal-organic frameworks (MOFs) are ideal platforms to realize exotic lattice models in condensed matter theory. In this work, we demonstrate the universal existence of topological corner states in 2D MOFs with a star lattice configuration, and confirm the intriguing higher-order nontrivial topology in the energy window between two Kagome-bands, or between Dirac-band and four-band. Furthermore, combining first-principles calculations and scanning tunneling microscopy measurements, the unique topological corner state is directly identified in monolayer $Ni_3(HITP)_2$ (HITP = 2,3,6,7,10,11-hexaimino-triphenylene) grown on the Au(111) substrate. Our results not only illustrate the first organic topological state in the experiments, but also offer an exciting opportunity to study higher-order topology in 2D MOFs with the large insulating band gap.

There are many exotic lattice models in condensed matter theory, ranging from single-particle[1,2] to many-body[3,4] cases. The studies of them not only give insight into the nature of topology, correlation and magnetization for quantum materials[5], but also promote the development of algorithms in computational physics[6]. However, the realization of them is a challenging task because solid-state materials found in nature often have the hybridized band structures[7], much more complex than the ideal models. In experiments, ultracold atoms[8] and photonic crystals[9] have been used to simulate certain lattice models, which can create artificial bands with tunable parameters, analogous to those formed by electrons in crystals. In addition to artificial systems, it is more urgent to develop suitable electronic platforms to explore these lattice models.

Metal-organic frameworks (MOFs) consist of metal atoms linked by organic ligands[10], forming a highly ordered porous network. This category of materials has attracted tremendous attention in chemical society, because of its excellent performance in gas storage and catalysis[11]. Due to the chemical flexibility for choosing metal and ligand fragments, its band structures around Fermi level can be easily tuned through custom designed molecular building blocks[12], showing an ideal electronic system to realize lattice models. Currently, Dirac and flat bands have been predicted in various two-dimensional (2D) MOFs[13], forming topological insulator (TI) and Chern insulator phases[14–16]. However, the tiny spin-orbital coupling (SOC) in organic materials makes it extremely difficult to detect topological boundary states within the SOC gap[17–19], greatly hindering the experimental progress in this field[20].

Recently, topological band theory has been extended from first-order to higher-order[21–24], where the nontrivial bulk topology of an $m$-dimensional $n$th-order TI is characterized by gapless states at ($m$-$n$)-dimensional boundary. Different from the conventional first-order TIs, the higher-order topology is protected by the crystalline[21] and chiral[25] symmetries, which doesn't originate from SOC. This indicates that the large insulating band gap in 2D MOFs has been overlooked in previous studies, which may be compatible with the second-order TI (SOTI) phase, facilitating the detection of in-gap topological boundary states. To date, the reported 2D SOTIs are mainly limited to some artificial structures[26–29] and a few inorganic solid-state materials[30–40], but seldom in MOFs.

In this work, based on tight-binding (TB) model, we first illustrate the universal existence of SOTI in 2D MOFs with the star lattice

[1]Hefei National Research Center for Physical Sciences at the Microscale, CAS Key Laboratory of Strongly-Coupled Quantum Matter Physics, Department of Physics, University of Science and Technology of China, Hefei, Anhui 230026, China. [2]Beijing National Laboratory for Condensed Matter Physics, Institute of Physics, Chinese Academy of Sciences, Beijing 100190, China. [3]Hefei National Laboratory, University of Science and Technology of China, Hefei, Anhui 230088, China. ✉e-mail: weihuawang@iphy.ac.cn; zfwang15@ustc.edu.cn

configuration and clarify the formation mechanism of topological corner states. Then, combining first-principles calculations and scanning tunneling microscopy (STM) measurements, we identify our proposal in monolayer $Ni_3(HITP)_2$ and confirm the realization of localized topological corner states within the nontrivial bulk band gap. Our results introduce a new platform to explore the organic higher-order topology, which is achievable by current experimental technology.

## Results

### Topological phase diagram

The 2D MOFs, constructed by $C_3$ symmetric cores, will form a hexagonal porous network, as shown in Fig. 1a. This is one most common structure synthesized in the experiments, having plenty of candidate materials[41,42]. This structure can also be seen as a star lattice if the lattice site is located at the vertex of each core, showing six sites per unit cell. The TB Hamiltonian of this lattice can be described by intracore (inter-core) hopping $t_1$ ($t_2$), as labeled in Fig. 1a (see also Supplementary Fig. 1). Depending on these two parameters, its band structures are divided into four types (Type-I to IV), as shown Fig. 1b. The Type-I (II) bands, satisfying $t_1 > 0$ ($t_1 < 0$) and $|t_2| > 1.5 |t_1|$, have two Kagome-bands with flat-band above (below) Dirac-band (Fig. 1c). The

Type-III (IV) bands, satisfying $t_1 > 0$ ($t_1 < 0$) and $|t_2| < 1.5 |t_1|$, have one Dirac-band and one four-band with four-band above (below) Dirac-band (Fig. 1d). The band gap between two groups of bands is closed down at $|t_2| = 1.5 |t_1|$, corresponding to the band transition point between Type-I (II) and III (IV).

To characterize the higher-order bulk topology of this lattice, its $Z_Q$ Berry phase ($\gamma$)[43] is calculated as a topological index (Supplementary Fig. 2). Physically, the Type-I and II bands (dominated by $t_2$) originate from an array of dimer states (solid line) in the decoupled limit of $t_1 = 0$ (dashed line), as shown in Fig. 1c. The lattice sites maintain $C_2$ symmetry, so the bulk topology in the band gap between two Kagome-bands is captured by $Z_2$ Berry phase. In contrast, the Type-III and IV bands (dominated by $t_1$) originate from an array of trimer states (solid line) in the decoupled limit of $t_2 = 0$ (dashed line), as shown in Fig. 1d. The lattice sites maintain $C_3$ symmetry, so the bulk topology in the band gap between Dirac-band and four-band is captured by $Z_3$ Berry phase. The calculated nonzero Berry phases for different bands are listed in Fig. 1b. For Type-I and II, the $Z_2$ Berry phase is $\gamma = \pi$. For Type-III and IV, the $Z_3$ Berry phase is $\gamma = 2\pi/3$ and $\gamma = 4\pi/3$, respectively. Thus, the Berry phase can be rewritten as $\gamma = 2\pi \cdot n$, where n = 1/2, 1/3, 2/3 is the filling factor of the Fermi-level for Type-I, II, III, IV bands, respectively. The quantized nontrivial Berry phase indicates that the ground

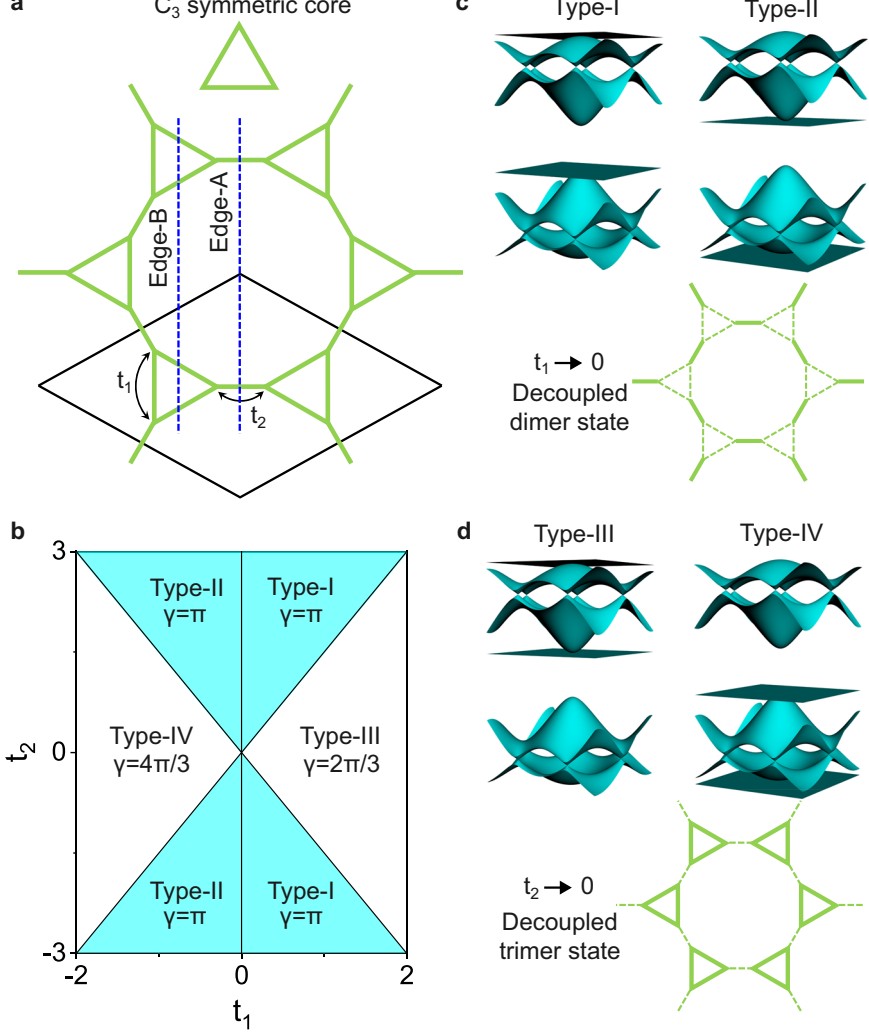

**Fig. 1 | Topological phase diagram and topological band structures of 2D MOF.** **a** Schematic 2D MOF assembled with the $C_3$ symmetric cores. The black solid-line denotes the unit-cell. The blue dashed-line denotes the termination of Edge-A or Edge-B. $t_{1,2}$ are two hopping parameters of the star lattice. **b** Topological phase diagram vs $t_{1,2}$, classified by Berry phase $\gamma$. Type-I, II, III, and IV denote four different bands. **c** Schematic Type-I and II bands, originating from the decoupled dimer states ($t_1 = 0$). **d** Schematic Type-III and IV bands, originating from the decoupled trimer states ($t_2 = 0$).

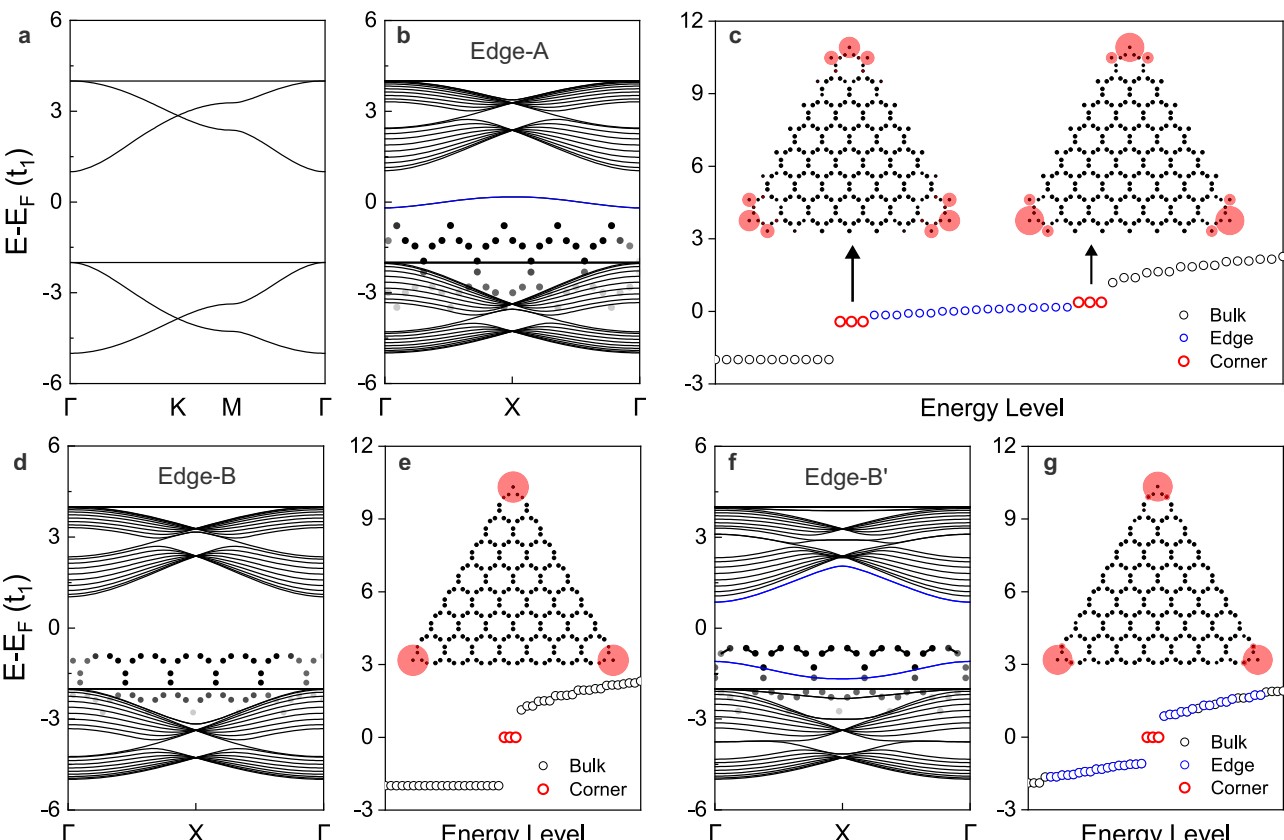

**Fig. 2 | Topological corner states in Type-I bands. a** Type-I bands with the Fermi-level between two Kagome-bands, corresponding to 1/2 filling. **b, d, f** Ribbon band structures with Edge-A, Edge-B and Edge-B′ termination, respectively. The inset shows the shape of the edge structure. Edge-B′ is a slightly modified Edge-B with $t_2 = 2.1t_1$ in the outmost dimers, as denoted by solid lines in the inset of **f**.

**c, e, g** Discrete energy-levels of triangular clusters with Edge-A, Edge-B, and Edge-B′ termination, respectively. The inset shows the spatial distribution of corner states. The circle size denotes the weighting factor of the corner states. The black, blue, and red colors in **b–g** denote the bulk, edge and corner states, respectively. The hopping parameter is set to $t_2 = 3t_1$.

state of the star lattice is adiabatically connected to the decoupled dimer or trimer states without breaking symmetry or closing the band gap[43], showing a stable 2D SOTI phase. Generally, band gap closing and reopening in topological systems are accompanied by a phase transition between trivial and nontrivial phases[44]. Here, the phase transition between Type-I (II) and III (IV) bands is described by two nontrivial phases with different topological indexes, exhibiting a new higher-order topological phase transition without the trivial phase[45].

## Topological corner states in Type-I bands

Besides the bulk topological index, the other smoking gun signature of 2D SOTI is characterized by topological corner states[30–40]. As shown in Fig. 2a, the Type-I bands consist of two Kagome-bands with the Fermi-level at 1/2 filling. The two typical edge terminations are labeled Edge-A and Edge-B (Fig. 1a), corresponding to the bond broken dimer and trimer, respectively. The ribbon band structures with Edge-A termination are shown in Fig. 2b. There is one edge state in the bulk band gap, which is detached from the valence and conduction bands. This edge state is induced by the coupled bond broken dimers along the edge (inset of Fig. 2b), forming a wire with one lattice site per unit cell. The discrete energy-levels of the triangular cluster with Edge-A termination are shown in Fig. 2c, having two groups of threefold degenerate corner states in the bulk band gap (above and below the edge states). Each corner state is localized at one corner region with a spatial distribution on three broken dimers (inset of Fig. 2c). The coupling among them will create three groups of corner states, but the zero energy state is hybridized with edge states, making it invisible (Supplementary Fig. 3). Hence, the Fermi-level lies in edge states without

the filling anomaly for corner states. Additionally, the ribbon band structures with Edge-B termination are also studied. Without the broken dimers along the edge, the edge state is eliminated from the bulk band gap, as shown in Fig. 2d. However, due to the broken dimer at the corner (inset of Fig. 2e), one group of spatially localized corner states is left in the discrete energy-levels of the triangular cluster, as shown in Fig. 2e. In this case, the Fermi-level lies exactly in corner states, holding a fractional charge of e/2 for each corner.

The appearance (absence) of the edge states with Edge-A (Edge-B) termination can be understood in an alternative way (Supplementary Fig. 4). In the decoupled limit of Edge-A termination, the edge, and bulk are constructed by monomer and dimer, respectively. In the decoupled limit of Edge-B termination, both the edge and bulk are constructed by dimer. Since the monomer and dimer have different energy levels, the edge and bulk states are distinguishable (indistinguishable) for ribbons with Edge-A (Edge-B) termination. To further support this analysis, we slightly decrease the hopping value in the outermost dimers ($t_2 = 2.1t_1$) of edge-B termination, making the edge dimer different from the bulk dimer[40], labeled Edge-B′ (inset of Fig. 2f). As expected, a pair of edge states appear in the bulk band gap, detached from the valance and conduction bands, as shown in Fig. 2f. The discrete energy-levels of the triangular cluster with Edge-B′ termination are shown in Fig. 2g, where the spatially localized one group of corner states keeps the same filling anomaly as that in Fig. 2e, but located between the gapped edge states. Different from the well-known mass inversion mechanism for creating the domain-wall states[36], the corner states here are created by bond broken dimers, which are insensitive to edge terminations.

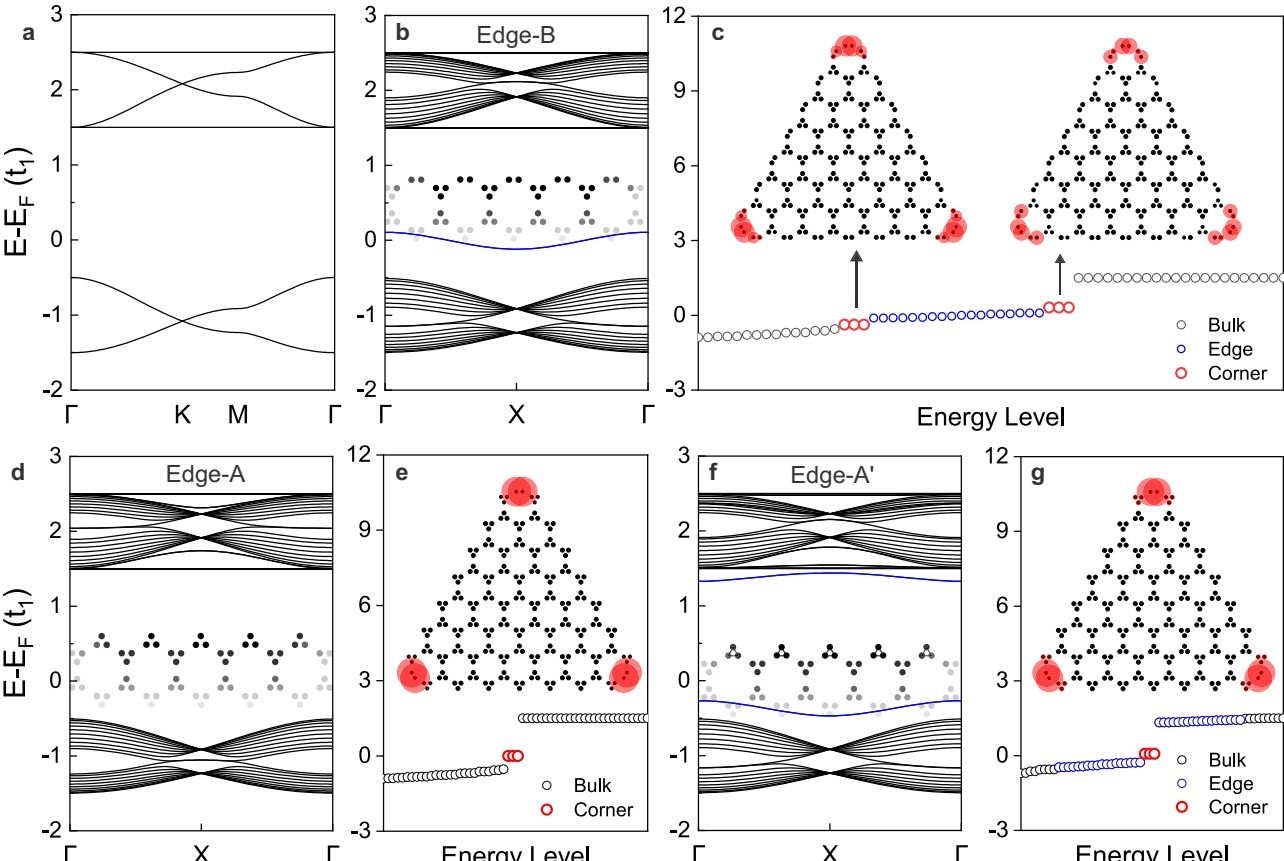

**Fig. 3 | Topological corner states in Type-III bands. a** Type-III bands with the Fermi-level between Dirac-band and four-band, corresponding to 1/3 filling. **b**, **d**, **f** Ribbon band structures with Edge-B, Edge-A and Edge-A′ termination, respectively. The inset shows the shape of the edge structure. Edge-A′ is a slightly modified Edge-A with $t_2 = 0.35t_1$ in the outmost trimers, as denoted by solid lines in the inset of **f**. **c**, **e**, **g** Discrete energy-levels of triangular clusters with Edge-B, Edge-A, and Edge-A′ termination, respectively. The inset shows the spatial distribution of corner states. The circle size and color have the same meaning as those in Fig. 2. The hopping parameter is set to $t_2 = 0.5t_1$.

## Topological corner states in Type-III bands

Similar corner states are observed in Type-III bands, consisting of one Dirac-band and one four-band with the Fermi-level at 1/3 filling, as shown in Fig. 3a. There is one edge state in the bulk band gap for the ribbon with Edge-B termination (Fig. 3b), which is created by bond broken trimers along the edge. The discrete energy-levels of the tri-angular cluster with Edge-B termination have two groups of corner states, siting above and below the edge states, as shown in Fig. 3c. For Edge-A termination with the intact trimer, the above edge state is eliminated from the bulk band gap (Fig. 3d). Meanwhile, one group of corner states is left in the discrete energy-levels of the triangular cluster (Fig. 3e and its inset), and each corner state at the Fermi-level holds a fractional charge of 2e/3. For Edge-A′ termination with a slightly decreased hopping value in the outmost trimers ($t_2 = 0.35t_1$) (inset of Fig. 3f), a pair of edge states are detached from the bulk states, as shown in Fig. 3f. The discrete energy-levels of the triangular cluster with Edge-A′ termination retain one group of corner states at the Fermi-level (Fig. 3g and its inset). Furthermore, the corner states in Type-II (IV) bands are similar to those in Type-I (III) bands (Supplementary Figs. 5, 6), which can hold a fractional charge of e/2 (e/3), showing the universal existence of corner states in 2D MOFs with the star lattice configuration.

## Type-I bands in Ni₃(HITP)₂

After illustrating the higher-order band topology in 2D MOFs from the theoretical aspects, its material realization is further investigated by first-principles calculations and STM measurements. We found that the

Type-I bands can be realized in monolayer Ni₃(HITP)₂ grown on Au(111) substrate. Through the reported on-surface coordination assembly method[46,47], monolayer Ni₃(HITP)₂ frameworks are synthesized on Au(111) substrate from Ni atoms and 2,3,6,7,10,11-Hexaamino-triphenylene (HATP) molecules under ultrahigh vacuum conditions. The high quality of the synthesized Ni₃(HITP)₂ frameworks are verified by STM image (Supplementary Fig. 7). Figure 4a shows the zoomed-in STM image of a triangular-shaped Ni₃(HITP)₂ framework with two edges forming a corner, which is overlapped with its atomic structures. The three-fold symmetric HITP molecules appear as two triangles sitting in a honeycomb lattice, and the coordinated Ni atoms are sand-wiched between every two adjacent HITP molecules (Supplementary Fig. 8). The Ni₃(HITP)₂ framework has a lattice constant of $2.17 \pm 0.03$ nm on Au(111) surface, corresponding to a $\sqrt{57} \times \sqrt{57}$ R6.6° unit cell. To reveal the electronic structures of the Ni₃(HITP)₂ frame-work, its density functional theory (DFT) bands are calculated. As shown in Fig. 4b, the calculated bands of Ni₃(HITP)₂ resemble the Type-I bands with a band gap of 0.41 eV. Based on these bands, the simulated STM image shows good agreement with the experimental data (Supplementary Fig. 8). The Wannier-bands (open circle in Fig. 4b) are also fitted to investigate the effective orbital of Type-I bands, where a star lattice is formed by six Wannier orbital centers (red circle) in each unit cell (inset of Fig. 4b). These first-principles results are consistent with our TB model calculations (Fig. 2), confirming the existence of non-trivial bulk bands in Ni₃(HITP)₂. Moreover, the similar Type-I bands are also observed in the projected bands of Ni₃(HITP)₂/Au(111) (Supplementary Fig. 9). Thus, the higher-order topology in Ni₃(HITP)₂ will be

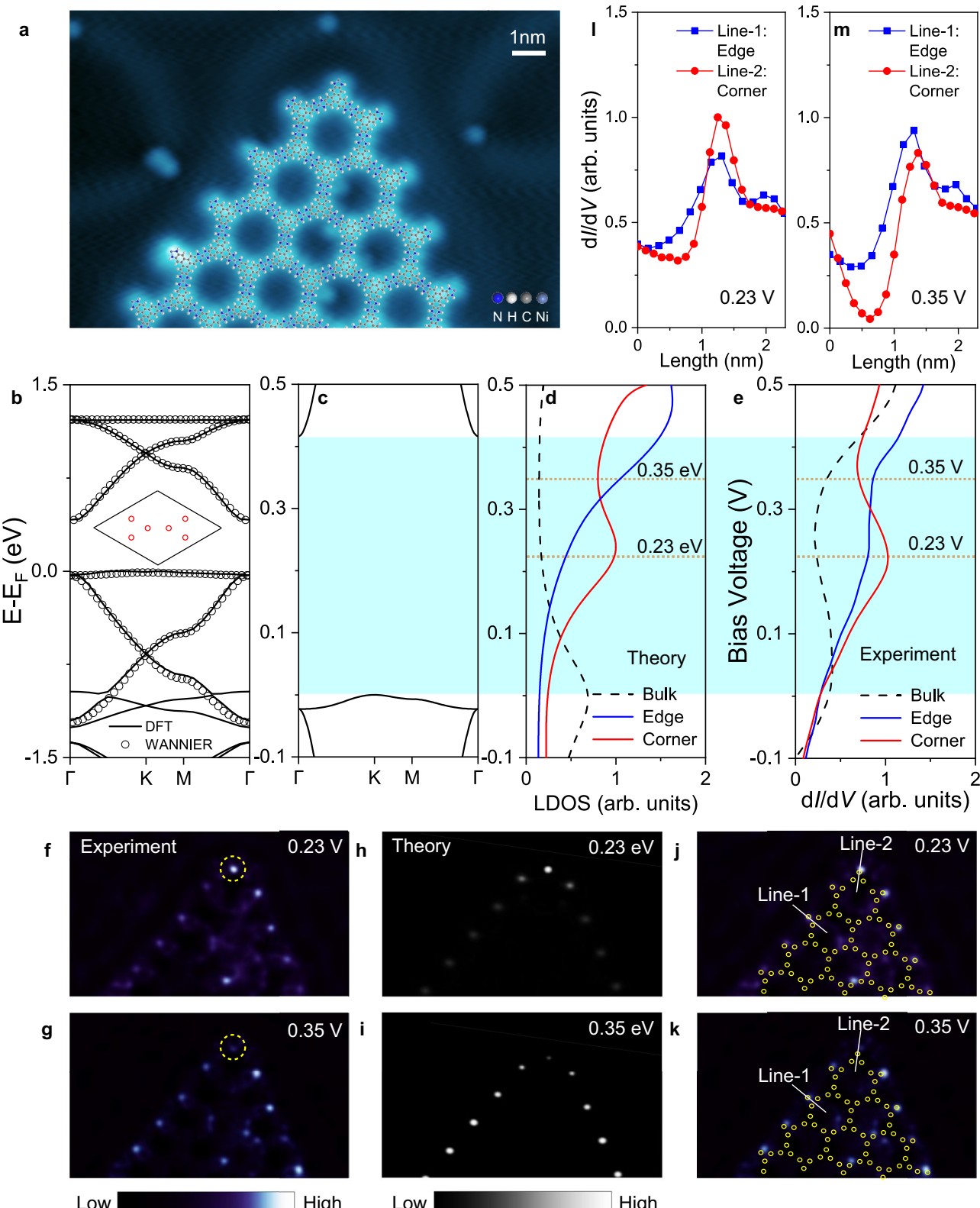

**Fig. 4 | Topological corner state comparison between theory and experiment in Ni₃(HITP)₂.** a STM topographic image of $Ni_3(HITP)_2$ triangular framework on Au(111) substrate (−0.3 V and 0.1 nA), which is overlapped with its atomic structures. **b** DFT bands of $Ni_3(HITP)_2$. The open circles are fitted Type-I Wannier-bands. The inset is fitted six Wannier orbital centers (red circles) in the unit cell. **c** Zoomed-in bands of **b** around the Fermi-Level. **d**, **e** Theoretical LDOS and experimental d$I$/d$V$ spectra of the bulk, edge and corner states in $Ni_3(HITP)_2$ triangular framework. **c**, **d**, **e** are aligned by setting Fermi-level and zero bias voltage together. The shadow color region denotes the bulk band gap. **f**, **g** d$I$/d$V$ maps at 0.23 V and 0.35 V, marked by two dotted lines in **e** showing the spatial distribution of experimental corner and edge states. The dashed yellow circle highlights the corner region. **h**, **i** LDOS maps at 0.23 eV and 0.35 eV, marked by two dotted lines in **d** showing the spatial distribution of theoretical corner and edge states. **j**, **k** are the same as **f**, **g** which are overlapped with the triangular cluster made of Wannier orbital centers (yellow circles). **l**, **m** d$I$/d$V$ profiles for edge and corner states at 0.23 V and 0.35 V, which are along line-1 and line-2 in **j**, **k**.

robust to the presence of substrate (Supplementary Figs. 10, 11), making it possible to detect boundary states in its large bulk band gap. Additionally, the band degeneracies at Dirac-point and quadratic-touching-point in $Ni_3(HITP)_2$ can be lifted by the SOC, realizing the 2D first-order TI in the tiny SOC gap[48]. However, the higher-order topology is not originated from the SOC, so it will also be robust to the presence of tiny SOC (Supplementary Fig. 12).

## Topological corner states in $Ni_3(HITP)_2$

To directly identify the topological corner states[49,50] in $Ni_3(HITP)_2$, we conduct current-imaging tunneling spectroscopy measurement on the $Ni_3(HITP)_2$ framework shown in Fig. 4a, and extract the differential conductance ($dI/dV$) spectra at the bulk (inner part of the framework), edge and corner sites. The representative spectra of bulk, edge and corner states are shown in Fig. 4e. By aligning the calculated Fermi-level and zero-bias voltage in experiments together, the theoretical bulk bands (Fig. 4c) and local density of states (LDOS) (Fig. 4d) are comparable with the experimental $dI/dV$ spectrum of the bulk state, where the similar curve shape and valence flat band peak are observed. In this way, the energy window of the bulk band gap in the $dI/dV$ spectrum is determined in the range of 0 to 0.41 V, as denoted by the shadow color region in Fig. 4c–e. Within the bulk band gap, the $dI/dV$ spectra have two significant features in Fig. 4e. First, the intensities of the edge and corner states are generally larger than that of the bulk state, showing the character of topological boundary states in the nontrivial bulk band gap. Second, the $dI/dV$ spectrum of the corner state has a peak (-0.23 V) in the middle of the bulk band gap, where its intensity is larger than that of the edge state, making the corner state detectable. Whereas at 0.35 V, the intensity of the edge state is larger than that of the corner state. All these features are consistent with the theoretically calculated LDOS for edge and corner states in the triangular cluster, as shown in Fig. 4d.

Furthermore, the $dI/dV$ maps are taken at 0.23 V and 0.35 V to visualize the spatial distribution of these states, as shown in Fig. 4f, g, respectively. At 0.23 V, the electronic states are mainly localized at the boundary of the triangular framework, and the corner region (dashed yellow circle) is brighter than the edge region (Fig. 4f), showing the character of the corner state. At 0.35 V, in contrast, the edge region becomes brighter than the corner region (Fig. 4g), showing the character of the edge state. In both maps, the bulk states are darker than the edge and corner states. The simulated LDOS maps at 0.23 eV (Fig. 4h) and 0.35 eV (Fig. 4i) show good agreement with the experimental maps. Moreover, the Wannier orbital centers (yellow circles) in the triangular cluster also match well with the real-space positions of the corner and edge states in $dI/dV$ maps at 0.23 V (Fig. 4j) and 0.35 V (Fig. 4k), illustrating the bond-broken dimer character for all of them. To quantitatively check the spatial localization of corner and edge states, the $dI/dV$ profiles are also taken at the edge (line-1) and corner (line-2), as marked by the white solid lines in Fig. 4j, k. Figure 4l, m show that both corner and edge states have a peak spectrum along line-1 and line-2 with the full-width at half maximum of ~6 Å, which are much localized states in real-space. Therefore, the localized topological corner state within the nontrivial bulk band gap is directly confirmed in $Ni_3(HITP)_2$, demonstrating the first experimental evidence of topological boundary states in 2D MOFs.

## Discussion

In summary, the universal higher-order topology is proposed theoretically in large gapped 2D MOFs, and its unique topological corner state is identified experimentally in 2D $Ni_3(HITP)_2$. Due to intrinsic merits of low cost, easy fabrication and mechanical flexibility for organic materials, the organic materials have been always matched with their inorganic counterparts to enable various cutting-edge research frontiers, such as organic superconductor, organic light emitting diode, organic solar cell, and organic field effect transistor.

The discovery of topological corner state in organic frameworks provides a new platform to explore the long-term dreamed organic topological states in experiments, greatly extending the category of conventional topological materials. The higher-order organic topological states can also be incorporated with magnetism, correlation and superconductivity to further investigate exotic quantum phenomena for application in both quantum computation and quantum simulation. We believe our work will inspire immediate research interest for studying higher-order topology in organic materials.

## Methods

### Tight-binding calculations

The tight-binding (TB) Hamiltonian of star lattice is written as

$$H = -t_1 \sum_{\langle i,j \rangle}^{intra-core} c_i^\dagger c_j - t_2 \sum_{\langle i,j \rangle}^{inter-core} c_i^\dagger c_j \tag{1}$$

where $c_i^\dagger$ ($c_i$) is the creation (annihilation) operator at lattice site $i = 1$ to 6 (Supplementary Fig. 1). $t_{1,2}$ is intra-core (inter-core) hopping parameter. In reciprocal space, the TB Hamiltonian is written as

$$H = \begin{pmatrix} 0 & -t_2 e^{i\mathbf{k}\cdot\mathbf{a}_2} & -t_1 & 0 & -t_1 & 0 \\ -t_2 e^{-i\mathbf{k}\cdot\mathbf{a}_2} & 0 & 0 & -t_1 & 0 & -t_1 \\ -t_1 & 0 & 0 & -t_2 e^{-i\mathbf{k}\cdot\mathbf{a}_1} & -t_1 & 0 \\ 0 & -t_1 & -t_2 e^{i\mathbf{k}\cdot\mathbf{a}_1} & 0 & 0 & -t_1 \\ -t_1 & 0 & -t_1 & 0 & 0 & -t_2 \\ 0 & -t_1 & 0 & -t_1 & -t_2 & 0 \end{pmatrix} \tag{2}$$

where $\mathbf{a}_{1,2}$ is lattice vector (Supplementary Fig. 1). The six eigenvalues are obtained as

$$E_{1,2} = -\frac{1}{2}\left(t_1 \pm \sqrt{9t_1^2 + 4t_2^2 + 4t_1 t_2 f(\mathbf{k})}\right)$$
$$E_{3,4} = -\frac{1}{2}\left(t_1 \pm \sqrt{9t_1^2 + 4t_2^2 - 4t_1 t_2 f(\mathbf{k})}\right) \tag{3}$$
$$E_{5,6} = t_1 \pm t_2$$

where $f(\mathbf{k}) = \sqrt{3 + 2\cos(\mathbf{k}\cdot\mathbf{a}_1) + 2\cos(\mathbf{k}\cdot\mathbf{a}_2) + 2\cos(\mathbf{k}\cdot\mathbf{a}_1 + \mathbf{k}\cdot\mathbf{a}_2)}$. Clearly, the band structures of star lattice are closely related to honeycomb and Kagome lattices. When $|t_2| > 1.5|t_1|$, its band structures exhibit a Kagome lattice feature, including two Kagome-bands. When $|t_2| < 1.5|t_1|$, its band structures exhibit a honeycomb lattice feature, including one Dirac-band and one four-band. When $|t_2| = 1.5|t_1|$, the band gap between two sets of bands is closed down.

The Berry phase of star lattice is calculated through the local twist method reported in previous works[43]. For $Z_2$ Berry phase, the local twist is introduced in one dimer between lattice site 1 and 2 (Supplementary Fig. 2a). The TB Hamiltonian includes bare and twisted two parts:

$$H(\theta) = -t_1 \sum_{\langle i,j \rangle \neq (1,2)}^{intra-core} c_i^\dagger c_j - t_2 \sum_{\langle i,j \rangle \neq (1,2)}^{inter-core} c_i^\dagger c_j - t_2 e^{i\theta} c_1^+ c_2 + h.c. \tag{4}$$

where $\theta_1 \in [0, 2\pi]$. Through exact diagonalization, the many-body ground state of $H(\theta)$ is obtained as $|\Phi_0(\theta)\rangle$. Then, $Z_2$ Berry phase is defined as a contour integral of Berry connection (Supplementary Fig. 2b):

$$\gamma = i \int_0^{2\pi} d\theta \cdot \langle \Phi_0(\theta) | \nabla_\theta \Phi_0(\theta) \rangle \pmod{2\pi} \tag{5}$$

Enforced by $C_2$ symmetry of TB Hamiltonian, $Z_2$ Berry phase is quantized to $n\pi$ with $n = 0, 1$. For $Z_3$ Berry phase, the local twist is

introduced in one trimer among lattice site 1, 2 and 3 (Supplementary Fig. 2c). The TB Hamiltonian also includes bare and twisted two parts:

$$H(\theta_1, \theta_2) = -t_1 \sum_{\langle i,j \rangle \neq (1,2,3)}^{\text{intra-core}} c_i^\dagger c_j - t_2 \sum_{\langle i,j \rangle \neq (1,2,3)}^{\text{inter-core}} c_i^\dagger c_j - t_1 e^{i\theta_2} c_1^+ c_2$$
$$-t_1 e^{-i(\theta_1+\theta_2)} c_2^+ c_3 - t_1 e^{i\theta_1} c_3^+ c_1 + \text{h.c}$$

(6)

where $\theta_{1,2} \in [0, 2\pi]$. Through exact diagonalization, the many-body ground state of $H(\theta_1, \theta_2)$ is obtained as $|\Phi_0(\theta_1, \theta_2)\rangle$. Then, $Z_3$ Berry phase is defined as a contour integral of Berry connection along path $I_{1,2,3}$ (Supplementary Fig. 2d):

$$\gamma_{1,2,3} = i \int_{I_{1,2,3}} d\vec{\theta} \cdot \langle \Phi_0(\theta_1, \theta_2) | \nabla_{\vec{\theta}} \Phi_0(\theta_1, \theta_2) \rangle \ (\text{mod } 2\pi)$$

(7)

The $C_3$ symmetry of TB Hamiltonian leads to $\gamma_1 = \gamma_2 = \gamma_3$, while the cancellation of path $I_{1,2,3}$ leads to $\gamma_1 + \gamma_2 + \gamma_3 = 0$, so $Z_3$ Berry phase is quantized to $\frac{2n\pi}{3}$ with $n = 0, 1, 2$.

To consider the effect of substrate induced small symmetry breaking perturbation, the random hopping ($\delta t$) and onsite energy ($\delta\varepsilon$) Hamiltonian is written as

$$H_{random} = \sum_i \delta\varepsilon_i c_i^\dagger c_i + \sum_{\langle i,j \rangle}^{\text{intra-core}} \delta t_{ij} c_i^\dagger c_j + \sum_{\langle i,j \rangle}^{\text{inter-core}} \delta t_{ij} c_i^\dagger c_j$$

(8)

To consider the effect of intrinsic SOC in star lattice[51], the SOC Hamiltonian is written as

$$H_{soc} = i\lambda \sum_{\langle\langle i,j \rangle\rangle} e_{ij}(c_{i\uparrow}^\dagger c_{j\uparrow} - c_{i\downarrow}^\dagger c_{j\downarrow})$$

(9)

where $\lambda$ is the intensity of intrinsic SOC, and $e_{ij} = +1(-1)$ denotes the right (left) turning electrons between the next-nearest-neighbor sites. Including the above small symmetry breaking perturbation and intrinsic SOC for type-I bands, the corresponding topological corner states (Supplementary Figs. 10, 12) and Berry phase (Supplementary Fig. 11) are almost the same as those shown in Figs. 1 and 2. Therefore, our proposed higher-order topology in the large band gap between two groups of Kagome-bands is very robust by considering the effect of substrate and SOC, making it detectable in the experiment.

## First-principles calculations

The first-principles calculations are carried out in framework of generalized gradient approximation with both hybrid B3LYP[52] and PBE[53] functional using Vienna Ab initio simulation package (VASP)[54]. B3LYP functional is used for free-standing monolayer $Ni_3(HITP)_2$, and PBE functional is used for $Ni_3(HITP)_2/Au(111)$. All calculations are performed with a plane-wave cutoff of 500 eV on $5 \times 5 \times 1$ Monkhorst-Pack k-point mesh.

The vacuum layer of 15 Å thick is used to ensure the decoupling between neighboring slabs. The substrate supercell is simulated by three layers of $\sqrt{57} \times \sqrt{57}R6.6°$ Au(111) structure. The DFT-D3 method[55] is used for van der Waals corrections. During structural relaxation, all atoms are relaxed until forces smaller than 0.01 eV/Å, where bottom-layer of Au atoms are fixed. The Wannier-bands and orbitals are fitted through Wannier90 package[56]. Based on Tersoff-Hamann approximation, the STM image is simulated for the occupied states within 0.9 eV below the Fermi level.

To simulate experimental d$I$/d$V$ maps, the theoretical local density of states (LDOS) for a triangular cluster are calculated by using fitted Wannier Hamiltonian. A small onsite energy $\varepsilon_{\text{onsite}}$ is added to lattice site at cluster boundary, namely, $\varepsilon_{\text{onsite}} = 0.05$ eV for trimer sites at corner, $\varepsilon_{\text{onsite}} = 0.08$ eV for edge sites nearest neighboring to trimer

sites at corner, and $\varepsilon_{\text{onsite}} = 0.16$ eV for the other edge sites. Since Wannier Hamiltonian is obtained by fitting the bulk bands, the addition of these extra onsite energies can better describe boundary effects in the experiments. The existence of topological corner state is robust to local perturbations. Then, LDOS is calculated as $\text{LDOS}(E, i) = \sum |\varphi_n(i)|^2 \delta(E - E_n)$, where $i$ denotes lattice site, $E_n$ and $\varphi_n$ denote the $n$th eigenvalue and eigenstate, and delta-function is approximated by a Lorentzian broadening of 0.2 eV. We consider $s$-orbital ($\varphi_s(r) = R_{10}(r)Y_{00}(\theta, \phi) \propto e^{-r/a}$ and $a = 5$ Å) as the basis for each lattice site, then, the theoretical LDOS maps at 0.23 eV and 0.35 eV are simulated by taking its value from the plane at 0.5 Å above triangular cluster.

## Growth of $Ni_3(HITP)_2$ on Au(111)

The atomically flat Au(111) substrate is cleaned by cycles of $Ar^+$ sputtering followed by annealing at 700 K for 30 min. The 2,3,6,7,10,11-Hexaaminotriphenylene (HATP) molecules are synthesized by previously reported method[57]. The HATP molecules are degassed in vacuum chamber at 500 K for removing HCl and then evaporated from molecular beam evaporator at approximately 573 K, while the neutral Ni atoms (99.99 + %, Goodfellow Cambridge Ltd.) are directly evaporated from electron beam evaporator. The high-quality monolayer $Ni_3(HITP)_2$ frameworks are assembled by sequentially co-depositing HATP molecules and Ni atoms on Au(111) substrate held at room temperature, and annealing the sample at 520 K for 30 min.

## Scanning tunneling microscope measurements

In ultra-high vacuum, the scanning tunneling microscope (STM) measurements are performed at 4.9 K with a base pressure better than $1 \times 10^{-10}$ mbar. The Pt/Ir tips are used in STM experiments. The bias voltage is applied to sample with respect to the tip. The differential conductance (d$I$/d$V$) signals are acquired using a lock-in amplifier with a sinusoidal modulation of 1517 Hz at 5 mV.

## Data availability

The data that support the findings of this study are available from the corresponding author upon reasonable request.

## Code availability

The codes that support the findings of this study are available from the corresponding author upon reasonable request.

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

## Acknowledgements

This work was supported by National Natural Science Foundation of China (No. 12174369, 11974402), Innovation Program for Quantum Science and Technology (No. 2021ZD0302800) and Fundamental Research Funds for the Central Universities.We thank Supercomputing

Center at USTC for providing computing resources. W.W. and W.Z. thank Prof. Long Chen for providing HATP molecules.

## Author contributions
Z.F.W. and W.W. conceived the project. T.H., T.Z., and Z.F.W. carried out theoretical calculations. W.Z. and W.W. carried out MOF growth and STS measurements. Z.F.W. prepared the manuscript with input from all authors.

## Competing interests
The authors declare no competing interests.
