## [Peer Review File · Nature Communications]

Identifying Topological Corner States in Two-Dimensional Metal-Organic FrameworksREVIEWER COMMENTS

Reviewer #1 (Remarks to the Author):

This study performed theoretical calculations to propose the unique higher-order topology in 2D MOF, and utilized STM technique to experimentally observe it in a well-known 2D conductive MOF, Ni₃(HITP)₂. In general, the idea is interesting, and the finding is indeed novel. The theoretical studies are solid and promising, and the high-quality STM data were collected. It is a very nice and solid work in the field of Physical Chemistry. However, I do not find the reason for publishing this work in Nature Communications. This work performed solid theoretical and experimental studies to investigate a unique fundamental phenomenon within a specific MOF, which is less likely to gain broad interest to the majority of all the Chemistry, Materials, Physics, and MOF communities. The material and fabricating technique are not new, and the STM study for MOF is not new as well. In addition, no potential applications were related, which also makes this work even less interesting to the Engineering community. Since Nature Communications is a Journal aiming for publishing multidisciplinary research that can provide high and broad impacts to cross-field researchers, I do not think this work is suitable, since the scope of this work is too narrow; it is even too narrow for a general chemistry Journal. I strongly suggest the authors to submit this paper to a high-quality Journal in the field of Physical Chemistry, such as Journal of Physical Chemistry Letters or Journal of Physical Chemistry C.

In addition to the suitability issue of this work for Nature Communications, I do not have much more comments and concerns on this work, since both the experimental and theoretical parts of this work are indeed solid. One minor comment is related to the experimental part for Ni₃(HITP)₂. The authors should provide the details regarding the synthesis of this 2D MOF, including the chemicals used here. Also, more characterization data (XRD data; isotherm of the powder grown together here) are needed to characterize this 2D MOF grown by the authors here in order to verify that the 2D MOF synthesized here is as pure as that reported in literatures.

Reviewer #2 (Remarks to the Author):

The authors demonstrate the existence of topological corner states in 2D MOFs with a star lattice configuration by using TB model, first-principles calculations, and scanning tunneling microscopy measurements. The Ni₃(HITP)₂ monolayer was grown on Au (111) substrate. The differential conductance (dI/dV) from the STM was compared to the LDOS from DFT to illustrate the existence of the corner and edge states in the MOF monolayer. The work topic is interesting and in time. The manuscript may be considered to accept after following problems are solved.

1) As introduced in the manuscript, the higher-order topology is protected by the crystalline and chiral symmetries. Can the authors tell readers what symmetries in the Ni₃(HITP)₂ monolayer protect the topological corner states? Why does the Au (111) substrate not break the higher-order topology of the MOF since the substrate may decrease the crystalline symmetries of the MOF monolayer?

2) It is well known that the differential conductance and the LDOS are two different physical quantities. The former is a nonequilibrium quantity (with electronic current flowing in the STM loop) while the latter is an equilibrium quantity. The conductance depends not only on the LDOS but also the transmission. Only in very small bias (<0.1V), we may take an approximation that the transmission is a constant in the transport energy window. Otherwise, it is not reasonable to take this approximation. The biases considered in Fig. 4 are pretty large.

3) Why are the DFT bands of the MOF in Fig. 4(b) and Fig. S9(b) not exactly the same?

4) The topological behaviors of the same monolayer ($\text{Ni}_3(\text{HITP})_2$) have been studied by Zhao et al in Phys. Rev. B 90, 201403(R) (2014). The authors should analyze whether the ground electronic states of the material they obtained are consistent with these reported in this literature.

5) There are some typos in the manuscript, such as in the abstract "with the large insulting gap" should be changed to "with the large insulating band gap". The manuscript should be checked.

Responses to Reviewer 1' comments:

Comment 1: This study performed theoretical calculations to propose the unique higher-order topology in 2D MOF, and utilized STM technique to experimentally observe it in a well-known 2D conductive MOF, $\text{Ni}_3(\text{HITP})_2$. In general, the idea is interesting, and the finding is indeed novel. The theoretical studies are solid and promising, and the high-quality STM data were collected. It is a very nice and solid work in the field of Physical Chemistry. However, I do not find the reason for publishing this work in Nature Communications. This work performed solid theoretical and experimental studies to investigate a unique fundamental phenomenon within a specific MOF, which is less likely to gain broad interest to the majority of all the Chemistry, Materials, Physics, and MOF communities. The material and fabricating technique are not new, and the STM study for MOF is not new as well. In addition, no potential applications were related, which also makes this work even less interesting to the Engineering community. Since Nature Communications is a Journal aiming for publishing multidisciplinary research that can provide high and broad impacts to cross-field researchers, I do not think this work is suitable, since the scope of this work is too narrow; it is even too narrow for a general chemistry Journal. I strongly suggest the authors to submit this paper to a high-quality Journal in the field of Physical Chemistry, such as Journal of Physical Chemistry Letters or Journal of Physical Chemistry C.

Reply: We sincerely thank the reviewer for reviewing our MS and considering it to be an interesting, novel and solid work. However, we don't agree with the reviewer's comments that the scope of our work is too narrow and less likely to gain broad interest from chemistry, materials and physics without potential applications. In the following part, we will further emphasize the importance and usefulness for realizing the higher-order topological corner states, and also outlook its impact for the on-going research of topological materials.

It is well-known that there is a general interest in the physical and material society for finding new topological states in new topological materials, which has been witnessed

by the development history of the topological physics. There are so many interesting quantum phenomena in topological states, which can only be detected after realizing them in realistic materials. Therefore, after the theoretical proposal of a new topological state in the models, the researchers will pay much attention to find suitable material systems to realize it and make experimental measurements. From gapped quantum spin Hall materials to 3D topological insulator materials, and then to gapless Weyl semimetal and nodal-line semimetal materials; from non-magnetic topological materials to quantum anomalous Hall materials and magnetic topological materials; from single particle topological materials to many-body correlated and superconducting topological materials, almost all studied topological materials follow this routine. The topological states in each class of topological materials have their unique topological physics, and the study of them will enrich the classification of topological states and topological materials. Based on this research background, the higher-order topological state was proposed in 2017. Then, the experimental works are mainly focused on the artificial structures for realizing it [Nat. Rev. Phys. 3, 520 (2021); Light: Sci. Appl. 9, 130 (2020)]. So far, the experimental realization of the higher-order topological states in realistic solid-state materials is still a challenging task. Recently, some progresses are reported for realizing the hinge states in 3D higher-order topological materials [Nat. Mater. 19, 974 (2020); Nat. Mater. 21, 1111 (2022)], but the experimental report is still lacking for the 2D higher-order topological states in natural solid-state materials. Since the 2D topological materials are attractive for device applications, it is emergent to develop a suitable platform for accelerating the experiment research in this field.

Due to the merits of low cost, easy fabrication and mechanical flexibility for the organic materials, during recent decades, many organic functional materials have successfully been matched with their inorganic counterparts to enable various cutting-edge research frontiers, such as the organic superconductor, organic light emitting diode, organic solar cell, and organic field effect transistor. The chemists and physicists have sustained great research interest in exploring the novel organic topological materials, which not only moves the organic materials into a new research area, but also offer a new route to

discover organic topological states. Although the researchers have predicted various organic topological states in organic materials via theoretical calculations, the experimental trials have not been successful and face many fundamental challenges. The most important issue is the weak spin-orbital coupling induced tiny nontrivial bulk gap, which prohibits the high-resolution detection of topological boundary states within this band gap. In our work, we found that the higher-order topological states can exist in the large-gapped 2D organic frameworks, showing the possibility to detect topological states in organic materials. Guided by our theoretical proposal, our STM experiments directly observed the localized topological corner state in the proposed nontrivial bulk gap, demonstrating the first experimental evidence for the higher-order topological state in 2D metal organic frameworks. Our work introduces an organic system with the large nontrivial gap to study the topological states, which is totally neglected in previous studies, and demonstrates the universal existence of higher-order topological state in the metal/covalent organic frameworks from both theoretical and experimental aspects. Therefore, our work confirms the realization of the higher-order topological state in the conventional large-gapped organic frameworks with the weak spin-orbital coupling, paving a new direction to study the topological states in organic topological materials for the on-going research.

Moreover, the topological corner state provides a prototype system to investigate many significant quantum phenomena in the condensed-matter physics. For example-1, in the field of the topological quantum computation, it is a long-term dream for people to manipulate and braid the Majorana fermion in topological superconductors for realizing quantum computing. Currently, the Majorana zero mode can be trapped at the magnetic field induced vortex center in surface states of topological superconductors. However, the magnetic field will also introduce extra trivial bound states in the vortex center, making it difficult to distinguish and obtain the pure Majorana zero mode for applications. So far, it is still a hard task to form an array of the uniform Majorana zero modes and braid them in the magnetic field. Since the Majorana zero mode in the vortex center is a 0D topological state, it is very similar to the 0D topological corner state

studied in this work. Physically, 2D second-order topological superconductor will automatically support the appearance of zero-energy Majorana corner mode, which can be realized by the proximity effect: putting the 2D (higher-order) topological insulator on the trivial superconductor [Phys. Rev. Lett. 121, 096803 (2018); Phys. Rev. B 100, 205406 (2019)]. Without the influence of magnetic field, the higher-order topological superconducting materials will be a suitable platform to construct the pure Majorana zero mode. Furthermore, utilizing the state-of-the-art lithographic etching technology, it is also possible to pattern the higher-order topological superconducting materials to a superstructure with uniform corners, which can generate an array of Majorana zero modes. Therefore, the realization of higher order topological state is useful for studying the Majorana zero mode in quantum computation.

For example-2, the topological corner state can also be linked with the spin/pseudospin degree of freedom. Namely, each corner state can be spin/pseudospin polarized, and different corners states can take the different spin/pseudospins [Phys. Rev. Lett. 128, 026801 (2022); Phys. Rev. Lett. 122, 086804 (2019)]. Since the topological corner state is robust to the perturbation of external environments, the spin/pseudospin in higher-order topological materials can serve as a topological memory to store “0” and “1” information. Moreover, due to the real-space coupling among different corner states, if the spin/pseudospin at one corner state is reversed, spins/pseudospins at its neighboring corner states will be reversed too for reducing the total energy of the system. In this way, a 1D wire structure made of such corner states is able to transfer the binary information with low dissipation, which is similar to the concept of quantum cellular automata [Science 311, 205 (2006)]. If an array of such corner states is constructed, it is also able to realize some functions of the quantum processor [Nature Commun. 13, 4483 (2022)]. Therefore, the realization of higher order topological state is useful for studying the spin/pseudospin related quantum information and transport.

For example-3, one knows that the lattice models have been intensively studied in the theoretical condensed-matter physics. Based on toy models, so many exotic quantum

phases have been predicted, such as the quantum spin liquid, Wigner crystal, exciton insulator, fractional quantum Hall and so on. However, the development of this field is greatly hindered without the experimental detections, so it becomes a critical issue to find a universal way for simulating these models in the experiments. Currently, the ultracold atoms and photonic crystals have been used to simulate certain lattice models. Unfortunately, there is no Fermi-level in these artificial systems, which makes it impossible to simulate the filling factor related quantum phases. Here, the topological corner state provides an alternative way to do such simulations. One corner state is an isolated energy level, and it is equivalent to a single lattice site. An array of the corner states is equivalent to a 2D lattice, where the lattice structures can be designed by the positions of corner states. In this way, different 2D lattice can be obtained by patterning the 2D higher-order topological materials into different structures. The hopping and interaction strength is tunable by lattice distance and substrate, and the filling factor is tunable by back gating. Therefore, the realization of higher order topological state is useful for studying the exotic lattice models in condensed-matter physics.

From the above discussions, one can see that the realization of higher order topological state is important and useful for both fundamental and application researches, and our work has significant impact for on-going research of topological materials, especially for the organic topological materials. It will draw broad research interest from chemistry, materials and physics. We hope the reviewer can be convinced by our explanations and suggest our revised MS for publication in Nature Communications.

Comment 2: In addition to the suitability issue of this work for Nature Communications, I do not have much more comments and concerns on this work, since both the experimental and theoretical parts of this work are indeed solid. One minor comment is related to the experimental part for $\text{Ni}_3(\text{HITP})_2$. The authors should provide the details regarding the synthesis of this 2D MOF, including the chemicals used here. Also, more characterization data (XRD data; isotherm of the powder grown together here) are needed to characterize this 2D MOF grown by the authors here in order to verify that

the 2D MOF synthesized here is as pure as that reported in literatures.

Reply: We thank the reviewer for considering our experimental and theoretical results are solid. We guess the reviewer has confused our MOF with the chemically synthesized MOFs in solutions. We should reiterate that the $\text{Ni}_3(\text{HITP})_2$ frameworks are synthesized through on-surface coordination assembly in ultrahigh vacuum conditions, following the reported recipe in Ref. 46. While the traditional hydro-/solvothermal methods often yield crystalline powders of MOFs [Energy Chem. 2, 100029 (2020)], the on-surface coordination assembly can synthesize single-layer MOFs without the contamination of solvent molecules. The detailed synthesis process has been provided in method section, and the chemicals used in the synthesis are Ni atoms and 2,3,6,7,10,11-Hexaaminotriphenylene (HATP) molecules.

Our synthesized $\text{Ni}_3(\text{HITP})_2$ frameworks are characterized by low-temperature STM, and the typical large-size and magnified images are shown in Supplementary Figs. 7-8. These images indicate the high quality of our samples are as good as those reported in Ref. 46 and in our recent work [Small 19, 2207877 (2023)]. Since our samples are prepared at the monolayer limit and kept in ultrahigh vacuum condition to avoid the contamination, we don't have other characterization data than STM.

Responses to Reviewer 2' comments:

Comment 1: The authors demonstrate the existence of topological corner states in 2D MOFs with a star lattice configuration by using TB model, first-principles calculations, and scanning tunneling microscopy measurements. The $\text{Ni}_3(\text{HITP})_2$ monolayer was grown on Au (111) substrate. The differential conductance (dI/dV) from the STM was compared to the LDOS from DFT to illustrate the existence of the corner and edge states in the MOF monolayer. The work topic is interesting and in time. The manuscript may be considered to accept after following problems are solved.

Reply: We sincerely thank the reviewer for reviewing our MS and recommending it for

publication in Nature Communications after we address some technique issues. We also thank the reviewer's constructive comments that help us further improve the quality of our MS.

Comment 2: As introduced in the manuscript, the higher-order topology is protected by the crystalline and chiral symmetries. Can the authors tell readers what symmetries in the $\text{Ni}_3(\text{HITP})_2$ monolayer protect the topological corner states? Why does the Au (111) substrate not break the higher-order topology of the MOF since the substrate may decrease the crystalline symmetries of the MOF monolayer?

Reply: We thank the reviewer's comments for helping us further clarify the symmetry effect. The band structures of monolayer $\text{Ni}_3(\text{HITP})_2$ have the Type-I bands, so its bulk topology is protected by C_2 symmetry and characterized by the nontrivial Z_2 Berry phase (Fig. 1b in main text). For the monolayer $\text{Ni}_3(\text{HITP})_2$ grown on Au(111) substrate, as the reviewer has mentioned, the exact C_2 symmetry will be broken. However, the optimized vertical distance between $\text{Ni}_3(\text{HITP})_2$ and substrate is as large as 3.21 Å, indicating a very weak-coupling between them. This is also confirmed by its projected band structures (Supplementary Fig. 9), showing the similar Type-I bands as the free-standing case. Moreover, our proposed higher-order topology is in a very large band gap between two groups of Kagome-bands. The substrate induced small symmetry breaking perturbation cannot change its nontrivial bulk topology, so our model with the exact symmetry is suitable to describe the essential physics. This is different to the spin-orbital coupling opened nontrivial band gap at Dirac-point or quadratic-touching-point. The first-order topology in a small spin-orbital coupling band gap may be destroyed by the symmetry breaking induced trivial gap.

To further support the above analysis, we use the weak random hopping/on-site energy to simulate the effect of C_2 symmetry breaking. The corresponding discrete energy levels of triangular clusters for Type-I bands are shown in Fig. R1. Compared to the results without perturbation (Fig. 2 in the main text), the same topological corner states

can be observed. Additionally, the value of the nontrivial Z_2 Berry phase only exhibits a small fluctuation under perturbation, which is approximately equal to the quantized value, as shown in Fig. R2. Therefore, due to the large nontrivial bulk band gap and the weakness of the coupling between substrate and $\text{Ni}_3(\text{HITP})_2$, our proposed topological corner states can still be observed in the STM experiment. We have added these new results as Supplementary Figs. 10-11 in the Supplementary Materials.

Figure R1. (a) and (b) Discrete energy-levels of triangular clusters for Type-I bands with Edge-A and Edge-B termination, respectively. The random hopping energy is added to t_1 and t_2 with $\delta t \in [-0.1t_1, 0.1t_1]$. (c) and (d) Discrete energy-levels of triangular clusters for Type-I bands with Edge-A and Edge-B termination, respectively. The random onsite energy is added with $\delta \varepsilon \in [-0.1t_1, 0.1t_1]$. The inset shows the spatial distribution of corner states. The circle size denotes the weighting factor of the corner states. The black, blue and red colors denote the bulk, edge and corner states, respectively. The hopping parameter is set to $t_2 = 3t_1$.

Figure R2. (a) Ideal topological phase diagram vs $t_{1,2}$ classified by Berry phase γ . Type-I, II, III, IV denote four different bands. (b) Berry phase γ along the dashed line in (a) by including random hopping energy $\delta t \in [-0.1, 0.1]$. (c) Berry phase γ along the dashed line in (a) by including random onsite energy $\delta \varepsilon \in [-0.1, 0.1]$. As a comparison, the ideal quantized Berry phase without perturbation is also plotted in (b) and (c).

Comment 3: It is well known that the differential conductance and the LDOS are two different physical quantities. The former is a nonequilibrium quantity (with electronic current flowing in the STM loop) while the latter is an equilibrium quantity. The conductance depends not only on the LDOS but also the transmission. Only in very small bias (< 0.1 V), we may take an approximation that the transmission is a constant in the transport energy window. Otherwise, it is not reasonable to take this approximation. The biases considered in Fig. 4 are pretty large.

Reply: We thank the reviewer for raising this critical issue. We agree with the reviewer that the STM differential conductance (dI/dV) depends on both LDOS and transmission, especially on the semiconductor surfaces [Surf. Sci. 181, 295 (1987); Surf. Sci. Rep. 26, 61 (1996)]. Nevertheless, on the metal substrates or insulating layers on metal substrates, the dI/dV spectra and dI/dV maps at bias voltages upto 2.0 V have been

widely used to reveal the distribution of LDOS [Science 302, 77 (2003); Science 304, 281 (2004)], showing the good agreement between experimental and theoretical results [Nature 531, 489 (2016); Nat. Commun. 7, 11507 (2016); Nat. Nanotechnol. 15, 22 (2020); Nat. Commun. 12, 5895 (2021); Nat. Commun. 13, 1705 (2022)]. The bias voltage considered in our work is in the range of -0.1 V to 0.5 V. Compared to the above STM experiments, we believe the differential conductance in this range can still provide a reasonable approximation to the LDOS of Ni₃(HITP)₂ sample.

Comment 4: Why are the DFT bands of the MOF in Fig. 4(b) and Fig. S9(b) not exactly the same?

Reply: We thank the reviewer for carefully reading our MS. The DFT bands in Fig. 4(b) is calculated by B3LYP functional, while the DFT bands in Fig. S9(b) is calculated by PBE functional. It is well-known that hybridized functional can give a more accurate band gap compared to the experimental value, so we choose the DFT bands with B3LYP functional in the main text, showing the good agreement between the theoretical and experimental results. In order to check the substrate effect for the band structures of free-standing Ni₃(HITP)₂, the three layers of Au(111) are used and the whole system includes over two hundred atoms. It is too expensive to calculate this large system with hybridized functional, so we only calculate the Ni₃(HITP)₂/Au(111) bands with PBE functional in Supplementary Materials. Although these two DFT bands are not exactly the same, they qualitatively coincide with each other, showing the similar Type-I bands. In the method section, the different DFT functional has been clarified.

Comment 5: The topological behaviors of the same monolayer (Ni₃(HITP)₂) have been studied by Zhao et al in Phys. Rev. B 90, 201403(R) (2014). The authors should analyze whether the ground electronic states of the material they obtained are consistent with these reported in this literature.

Reply: We thank the reviewer for letting us know this interesting work and we have

cited it in our revised MS as Ref. 48. The ground electronic states of our calculations are the same as those reported in the reviewer mentioned PRB. The PBE bands in Fig. 2(a) of this PRB is the same to our calculated PBE bands in Supplementary Fig. 9(b). One can see two groups of Kagome-bands in both of them, sitting above and below the Fermi-level, showing the character of Type-I bands proposed in our work. However, the spin-orbital coupling is considered in the PRB work but neglected in our work, which lift the band degeneracies at Dirac-point (K point) and quadratic-touching-point (Γ point), realizing the first-order topological insulator in these spin-orbital coupling gaps. In the PRB work, the band gap between two groups of Kagome-bands is trivial in the definition of the first-order topological insulator, so it has not been studied. In our work, we found this band gap is nontrivial for the definition of the second-order topological insulator, which can support the appearance of the topological corner states. As a further confirmation of our results, we also consider the effect of intrinsic spin-orbital coupling in our tight-binding model. As shown in Fig. R3, besides the gap opening at K point and Γ point, the topological corner states are the same as those without spin-orbital coupling (Fig. 2 in the main text). Therefore, the spin-orbital coupling doesn't change our proposed higher-order topology. Our work and the PRB work study the $\text{Ni}_3(\text{HITP})_2$ with the same electronic states, but focus on different band gap and different topological physics. We have added these new results as Supplementary Fig. 12 in the Supplementary Materials.

Figure R3. (a) Type-I bands with intrinsic SOC. (b) and (d) Ribbon band structures with Edge-B and Edge-A termination, respectively. The insets show the shape of the edge termination. (c) and (e) Discrete energy-levels of the triangular cluster with Edge-B and Edge-A termination, respectively. The insets show the spatial distribution of corner states. The circle size represents the weighting factor of the corner states. The black, blue and red colors in (b-e) denote the bulk, edge and corner states, respectively. The hopping parameter is set to $t_2=3t_1$, and the intensity of SOC is set to $\lambda=0.1t_1$. Including the intrinsic SOC, the number of the corner states is doubled, but the overall feature is the same to that without SOC.

Comment 6: There are some typos in the manuscript, such as in the abstract “with the large insulting gap” should be changed to “with the large insulating band gap”. The manuscript should be checked.

Reply: We thank the reviewer’s comments. We have carefully rechecked our MS and corrected all typos we found.

REVIEWERS' COMMENTS

Reviewer #1 (Remarks to the Author):

The authors have provided detailed responses regarding the concerns raised previously, and the responses are convincing. Thus, I can recommend the publication of this work.

Reviewer #2 (Remarks to the Author):

I am satisfied with the authors' replies. They have made substantial improvements to the manuscript. I recommend accepting this manuscript.

Responses to Reviewer 1' comments:

Comment 1: The authors have provided detailed responses regarding the concerns raised previously, and the responses are convincing. Thus, I can recommend the publication of this work.

Reply: We sincerely thank the reviewer for recommending the publication of our work.

Responses to Reviewer 2' comments:

Comment 1: I am satisfied with the authors' replies. They have made substantial improvements to the manuscript. I recommend accepting this manuscript.

Reply: We sincerely thank the reviewer for recommending accepting our manuscript.